# Satisfaction with Urinary Incontinence Treatments in Patients with Chronic Spinal Cord Injury

**DOI:** 10.3390/jcm11195864

**Published:** 2022-10-04

**Authors:** Sheng-Fu Chen, Yu Khun Lee, Hann-Chorng Kuo

**Affiliations:** Department of Urology, Hualien Tzu Chi Hospital, Buddhist Tzu Chi Medical Foundation, Buddhist Tzu Chi University, Hualien 970, Taiwan

**Keywords:** neurogenic lower urinary tract dysfunction, bladder management, urinary incontinence, surgery, spinal cord injury

## Abstract

Purpose: To investigate the long-term satisfaction and complications in chronic spinal cord injury (SCI) patients after various bladder management strategies and surgical procedures for the treatment of urinary incontinence. Methods: Patients at a single institution with chronic SCI who received bladder management treatment or surgical procedure to improve urinary continence were retrospectively assessed. Thorough urological examinations and videourodynamic studies were performed. Patients were treated either through conservative approaches including medical treatment, clean intermittent catheterization (CIC), cystostomy, and indwelling urethral catheter, or through surgical procedures including detrusor botulinum toxin (Botox) injections, augmentation, ileal conduit, Kock pouch diversion, continent cystostomy, suburethral sling, and artificial urethral sphincter (AUS) implantation. The patients’ satisfaction with urinary continence improvement, causes of dissatisfaction, long-term complications, and overall satisfaction with bladder and voiding condition were assessed. Results: A total of 700 consecutive patients were enrolled in this study. High satisfaction rates were noted after detrusor Botox injection (81.1%), augmentation enterocystoplasty (91.4%), autoaugmentation (80%), Kock pouch diversion, and continent cystostomy (all 100%). Fair satisfaction rates were noted after ileal conduit diversion (66.7%), suburethral sling (64.3%), and AUS implantation (66.7%). Patients who received conservative treatment with medicines, CIC, cystostomy, or an indwelling urethral catheter all had less-satisfactory outcomes (all < 40%). Conclusion: Overall satisfaction with surgical procedures aimed to improve urinary continence in chronic SCI patients was higher than with conservative bladder management (35.4%). Appropriate surgical procedures for chronic SCI patients with neurogenic lower urinary tract dysfunction (NLUTD) and urological complications yielded satisfaction with both urinary continence improvement and with overall bladder and voiding condition.

## 1. Introduction

Neurogenic lower urinary tract dysfunction (NLUTD) is an umbrella term for dysfunctions of the urinary bladder and urethra resulting from damage to the peripheral or central nervous system (CNS). NLUTD in spinal cord injury (SCI) remains a challenging urological disorder. SCI patients can lose the ability to store urine due to neurogenic detrusor overactivity (NDO) or urethral sphincter incompetence. They may lose the ability to empty the bladder due to detrusor areflexia (DA), detrusor underactivity (DU), bladder neck dysfunction (BND), or detrusor sphincter dyssynergia (DSD), or they may suffer a combination of storage and emptying dysfunctions due to either DSD or detrusor hyperreflexia along with inadequate contractility. In addition, impaired bladder compliance may determine upper urinary tract deterioration. NLUTD places a considerable disease burden on SCI patients and adversely affects their quality of life (QoL) [1].

The major goals of NLUTD treatment include the preservation of renal function, the prevention of recurrent urinary tract infection (UTI), and the reduction of urinary incontinence, with the achievement of which addressing the secondary goal of improving QoL [2]. Treatment strategies should be based on the patient’s symptomatology, urodynamic data, renal function, and upper tract imaging. Treatment should be individualized according to the severity of the disability, the patient’s mental and physical condition, and the specific urinary tract dysfunction(s) [3]. Lower urinary tract symptoms often seen in SCI patients include urgency and urgency urinary incontinence (UUI). Patients with BND or DSD may also experience difficulty emptying the bladder and with urinary retention [4].

In patients with SCI above the T8 level, bladder overdistention, stool impaction, or UTI can trigger autonomic dysreflexia (AD) [5]. In those with a complete spinal cord lesion at the T6-S2 levels, involuntary bladder contractions without sensation and DSD usually develop [4]. When spinal cord lesions are below S2, DA with retention of residual urethral sphincter tone may cause difficulty with urination. NDO and DSD commonly occur in patients with suprasacral cord lesions, and are associated with increased intravesical pressure and upper tract deterioration [6]. Patients with DSD usually suffer from urinary incontinence and a large post-void residual volume requiring clean intermittent catheterization (CIC) or an indwelling Foley catheter [4]. These urological complications may contribute to reduced QoL and can lead to more serious disorders such as AD, UTI, and upper urinary tract deterioration [7]. While chronic SCI patients with urinary incontinence are usually very keen to attain continence, they are not always happy with the storage and voiding conditions that result from treatment. This study aimed to investigate the long-term satisfaction of chronic SCI patients following different types of incontinence treatment and the complications that occur with each treatment type.

## 2. Materials and Methods

Patients with chronic SCI who received treatment for urinary incontinence at our institute were retrospectively studied. The study was approved by Tzu Chi General Hospital (IRB 110-033-B). Most of the patients in our sample had been referred to our institute from across Taiwan and had had a history of SCI of >3 years when enrolled in the study. Each patient underwent a routine urological workup comprised of urinalysis, urine culture, renal function tests (renal sonography, serum levels of blood urea nitrogen, creatinine, electrolytes, and estimated glomerular filtration rate), and cystoscopy. Each also underwent a videourodynamic study (VUDS) to evaluate their detrusor contractility, their bladder neck and external sphincter coordination during urination, the presence of low bladder compliance or vesicoureteral reflux (VUR), and their ability to instigate spontaneous voiding.

After these thorough clinical examinations, patients were questioned about their current bladder condition and bladder outlet dysfunction. Either a surgical procedure or conservative treatment was then recommended based on the priorities observed in the management of NLUTD in SCI patients. The priority of conservative versus surgical management was based on the patients’ symptoms, upper tract conditions, and expectations. The procedures of bladder management were thoroughly discussed with the patients and all patients were fully informed about the advantages and potential adverse effects of the procedure. These are, in order of importance, as follows: (1) the preservation of renal function, (2) freedom from urinary tract infections, (3) efficient bladder emptying, (4) freedom from indwelling catheters, (5) patient agreement with the treatment approach, and (6) avoidance of long-term medication requirements after treatment [7]. Patients with upper urinary tract deterioration, recurrent pyelonephritis, high-grade VUR, bladder contraction with high intravesical pressure, or severe urinary incontinence, with or without DSD, were treated with the procedures recommended by the current guidelines [8]. Patients could choose between treatments that would allow either urinary continence or spontaneous voiding with urinary incontinence, with or without a catheter. Choices were discussed with patients, taking into account their preferences, self-care ability, available facilities, caregiving assistance, and family support.

After their initial treatment, patients were regularly followed up with at our outpatient clinic. Their renal function, bladder storage condition, and voiding efficiency were assessed by urodynamic study, and the tests were performed either every 6 months or annually depending on the patient’s upper urinary tract condition. If the patient was dissatisfied with their current lower urinary tract status or the occurrence of urological complications or if they wished to change treatments, additional surgical procedures or conservative treatments were performed. Patients with NDO, AD, BND, DSD, low bladder compliance, or the presence of VUR received repeated VUDS every year to assess the lower and upper urinary tract conditions, and any additional procedures or medications were given to protect renal function, prevent UTI, and improve urinary function. Chart review and telephone interviews were performed to assess the patients’ satisfaction with urinary continence, any causes of dissatisfaction or long-term complications after treatment, their overall satisfaction with their final bladder and voiding status, and to determine whether there had been any treatment changes.

Patient satisfaction with improvements in urinary continence as well as voiding condition was assessed using a global response assessment (GRA) with rating choices ranging from −3 to +3, to denote responses from “markedly worse” to “markedly improved”. A GRA score of +2 or +3 was considered a satisfactory outcome. Patients might have had improvement of urinary continence after treatment but were bothered by bladder emptying and thus rated their outcomes as unsatisfactory. The GRA for each treatment was analyzed among SCI patients with different injury levels and different VUDS findings. Overall satisfaction with current bladder and voiding status was graded on a four-point scale as either “very satisfied”, “satisfied but wish to change”, “acceptable and no need to change”, or “not satisfied at all”. Chi-square tests were performed to analyze the satisfaction rates of different subgroups for each treatment. The complication rates and treatment changes were recorded based on observational data. Post-treatment complication was defined by having a symptom or urinary tract condition that was more severe than baseline, such as urinary incontinence, difficulty with urination, recurrent UTI, exacerbated hydronephrosis, or AD. All calculations were performed using SPSS Statistics for Windows, version 20.0 (IBM Corp., Armonk, NY, USA). *p*-values < 0.05 were considered statistically significant.

## 3. Results

A total of 700 consecutive patients were enrolled in this study, including 511 male and 189 female patients. The mean age at which spinal injury occurred was 35.7 ± 17.6 (range 1–89) years, and the duration of the SCI follow-up was 13 ± 6.8 (range 3–25) years. The SCI was cervical in 238 (34%) patients, thoracic in 337 (48.1%), lumbar in 99 (14.1%), and sacral or infrasacral in 26 (3.7%). The initial VUDS examination identified NDO without DSD in 52 (7.4%) patients, NDO with DSD in 375 (53.6%), NDO with AD and DSD in 85 (12.1%), DA or DU in 162 (23.1%), and DA/DU with intrinsic sphincter deficiency (ISD) in 26 (3.7%). During the follow-up period, 77 (11.0%) patients died because of urological complications or other diseases. Patients’ bladder and voiding status were assessed based on their final chart record or interview. Table 1 shows the baseline patient demographics of all patients with different levels of SCI.

Patients’ satisfaction ratings with post-treatment urinary continence revealed high satisfaction rates after detrusor botulinum toxin A (Botox) injections (81.1%), augmentation enterocystoplasty (91.4%), auto-augmentation (80%), Kock pouch diversion (100%), and continent cystostomy (100%). Fair satisfaction rates were noted after ileal conduit diversions (66.7%), suburethral slings (64.3%), and artificial urethral sphincter (AUS) implantations (66.7%). Patients who received conservative treatment with medicines, CIC, cystostomy, or indwelling urethral catheters reported less-satisfactory outcomes (all < 40%) (Table 2).

When we assessed the satisfaction rates according to different VUDS findings, we found high satisfaction rates only in patients with NDO and without DSD (72%) treated with antimuscarinic agents and in those with NDO (all > 80%), but not in those with DA/DU with low compliance after detrusor Botox injections (Table 3). Most patients reported a high satisfaction rate after augmentation enterocystoplasty, auto-augmentation, ileal conduit diversion, and continent cystostomy. The patients who chose CIC, cystostomies, and indwelling urethral catheters did not report a satisfactory outcome in any VUDS subgroup.

Although treatments were performed to eradicate urological complications, facilitate bladder storage, and improve QoL, the long-term dissatisfaction and complication rates remained high, including recurrent UTI (*n* = 513, 73.3%), difficult urination (*n* = 153, 21.9%), persistent hydronephrosis (*n* = 18, 2.6%), persistent urinary incontinence (including UUI and stress urinary incontinence (SUI)) (*n* = 150, 21.4%), and persistent AD (*n* = 91, 13%) (Table 4).

After initial treatment, a high percentage of patients chose to change to a different form of treatment. Eighteen patients who received detrusor Botox injections went on to undergo augmentation enterocystoplasty (*n* = 16, 8.5%), auto-augmentation (*n* = 1, 0.5%), or ileal conduit diversion (*n* = 1, 0.5%) for intractable urinary incontinence and upper urinary tract deterioration. Eleven (10.9%) patients initially treated with augmentation enterocystoplasty went on to receive detrusor Botox injections for persistent AD or UUI, while 18 switched to cystostomy (*n* = 9, 8.9%) or indwelling urethral Foley catheter (*n* = 9, 8.9%) because of difficulty performing self-CIC. During the follow-up period, 101 (14.4%) patients underwent further surgical procedures to facilitate spontaneous voiding, including urethral Botox injections (*n* = 44), TUI-BN (*n* = 26), TUIP or TURP (*n* = 19), and external sphincterotomy (*n* = 12).

Table 5 shows overall satisfaction rates with final bladder and voiding status after treatment for urinary incontinence. These satisfaction rates were slightly lower than those for urinary continence improvement in patients who received detrusor Botox injections (from 81.1% to 67.7%) and augmentation enterocystoplasty (from 91.4% to 77.2%). However, satisfaction was higher in patients who received CIC (from 30.9% to 60.8%) and suburethral slings (from 64.3% to 78.6%).

The results of this study revealed an overall satisfaction rate with surgical procedures (271/319, 85.5%) to improve urinary continence in chronic SCI patients that was higher than that for conservative treatments (135/381, 35.4%). Overall satisfaction with bladder and voiding status after surgical procedure (228/319, 71.5%) and conservative treatment (110/381, 28.9%) were slightly lower because some patients were dissatisfied with the outcome of their initial treatment. Chronic SCI patients who chose surgical procedures to improve urinary continence had significantly higher satisfaction with both urinary continence and overall bladder status after treatment.

## 4. Discussion

Chronic SCI patients with contracted bladders, obstructive uropathy, and recurrent UTIs are usually prepared to undergo surgical procedures and subsequent CIC to treat complications and provide urinary continence. However, patients without urological complications, upper tract deterioration, or severe urinary incontinence are more likely to choose conservative treatments such as CIC, cystostomies, and indwelling urethral catheters. However, the results of this study suggest that urological surgeries not only eradicate urological complications, but also provide a better QoL and greater improvement of urinary continence.

Conservative management is the mainstay of urological treatment for NLUTD if patients cannot empty their bladders efficiently. Patients may be instructed to use abdominal stimulation, the Crede maneuver, or abdominal Valsalva straining to trigger reflex voiding, or to use CIC either by themselves or with the help of a caregiver [9,10]. However, spontaneous voiding by reflex or abdominal straining can endanger the upper urinary tract, except in select SCI patients with regular urological follow-up [11]. Using these approaches can lead to long-term urological complications and exacerbate LUTD such as recurrent UTIs, hydronephrosis, urinary incontinence, or AD, and these patients often request alternative treatment to attain normal urinary tract structure and function [12].

Long-term indwelling catheters should be avoided in chronic SCI patients. However, patients with complete tetraplegic SCI usually have poor hand function and urinary incontinence. If CIC is not feasible, an indwelling urethral catheter or suprapubic cystostomy are viable alternatives [13]. Most chronic SCI patients have both bladder storage and voiding problems, but not all are willing to undergo surgical procedures or minimally invasive treatments, such as detrusor Botox injections with subsequent CIC, to improve urinary continence. Rather, they may be more inclined to choose reflex voiding, CIC alone, cystostomy, or an indwelling urethral catheter for partial relief of urinary incontinence, especially for tetraplegic patients [14]. Although the satisfaction rate with urinary continence improvement was low, most chose conservative treatment, possibly because this approach can potentially solve both storage and emptying problems at the same time. For this reason, many SCI patients with a disease duration of more than 5 years switched from CISC/CIC to an indwelling urethral catheter or cystostomy [15]. Therefore, an individualized, patient-tailored approach is required for the management of NLUTD associated with neurological disorders [16]. This study also showed that, although CIC or indwelling catheters are simple and non-invasive procedures, a considerable portion of patients experience long-term complication, which ultimately results in low satisfaction with their overall bladder condition.

Anticholinergics are recommended by the European Association of Urology [17] and a UK consensus group [18] for urinary incontinence and protection of renal function in SCI patients. However, a study in Taiwan found that only 18% of patients stayed on pharmacotherapy without additional urinary catheterization [19]. In this study, we found drug therapy was only effective in improving of urinary continence in SCI patients with NDO and without DSD or ISD. The overall satisfaction rate with final bladder and voiding status was lower than 50%, implying that pharmacotherapy is insufficient as a single or primary therapeutic approach in chronic SCI patients with urinary incontinence.

The clinical application of Botox was first introduced in urology for the treatment of NLUTD in chronic SCI patients [20]. Detrusor Botox injections have been reported to provide satisfactory results in the reduction of DO and intravesical pressure, and to improve urinary incontinence in SCI patients with NDO [21,22]. DU develops after detrusor Botox injections, and improvements in the urodynamic and QoL parameters last for around 9 months [23]. Detrusor Botox injections can also decrease the occurrence of AD in SCI patients, even after augmentation enterocystoplasty. In addition, repeated detrusor Botox injections can also reduce UTI episodes and preserve renal function [24].

Although repeated Botox injections are necessary to achieve the desired effects, most patients are willing to tolerate this in exchange for a better QoL [25]. We found a high satisfaction rate with urinary continence improvement after detrusor Botox injections in all suprasacral SCI patients with NDO with or without DSD. However, recurrent UTIs, difficult urination, persistent UUI/SUI, and the need for CIC reduced long-term satisfaction. Previous studies have found that 200 U Botox injections every 6 months for NDO in chronic SCI patients provide a satisfactory initial outcome; however, only 20% of patients continue with this treatment [26].

Bladder augmentation using a segment of the intestine [27] and auto-augmentation using myomectomy [28] can both prevent bladder reflux and provide a bladder with low intravesical pressure and a large capacity. However, long-term complications remain a problem. These may include stone formation, loose stools, metabolic acidosis, and chronic UTIs [29]. In this study, augmentation enterocystoplasty, autoaugmentation, ileal conduit diversions, Kock pouch diversions, and continent cystostomies were performed in cases with severely contracted bladders, VUR, hydronephrosis, recurrent UTIs, and AD. These surgical procedures relieved not only urinary incontinence, but also urological complications; therefore, the satisfaction rates were high for both urinary continence improvement and overall bladder and voiding status. A comparison of patients’ QoL rates after treatment with augmentation enterocystoplasty and repeat detrusor Botox injections showed that the former results in a higher satisfaction rate, possibly because Botox requires repeat injections and has relatively high complication rates [30]. Although recurrent UTIs and persistent incontinence still bothered some patients after augmentation, they were usually willing to accept this, along with the need for CIC [29]. SCI patients may consider augmentation enterocystoplasty to obtain life-long therapeutic effects rather than periodic Botox-A injections, especially when there are adverse effects after each Botox injection.

For SCI patients with severely contracted bladders and urethral incompetence, a continent lower urinary tract reconstruction such as a Kock pouch or closure of the bladder neck, plus continent ileostomy and bladder augmentation, can offer the opportunity to attain urinary continence while evacuating the bladder by CIC from an ileostoma [31]. When the bladder capacity and intravesical pressure are sufficient for CIC, the suburethral sling procedure or an AUS implantation with subsequent CIC are feasible means of creating a competent bladder outlet. The results of this study revealed that a fair satisfaction rate can be achieved with this approach, but persistent urinary incontinence can occur because of overflow incontinence without bladder sensation, due to low bladder compliance. Concomitant treatments to reduce intravesical pressure and increase bladder capacity are necessary for these patients.

After the initial procedure for the treatment of urinary incontinence, SCI patients often underwent further procedures or changed treatment. Detrusor Botox injections could not provide complete continence and resolution of AD or increased bladder capacity. Therefore, 18 (9.6%) patients underwent bladder augmentation or diversion, and 29 (15.6%) shifted to CIC or an indwelling catheter after initial treatment with Botox. Eighteen (17.8%) patients who initially underwent bladder augmentation changed to an indwelling catheter because of difficulty performing CIC. Although some SCI patients were satisfied with their first treatment choice, some still expected to attain spontaneous voiding ability. Therefore, during the follow-up period, 101 (14.4%) patients underwent further surgical procedures to facilitate spontaneous voiding, indicating that spontaneous voiding is important to chronic SCI patients treated for urinary incontinence.

Appropriate management of NLUTD in patients with SCI is an art and a challenge for urologists. Most patients with suprasacral SCI had NDO with or without DSD, and most also had both bladder storage and emptying dysfunctions [6]. The presence or absence of risk factors for renal damage and symptomatic UTI affects the treatment options, including the choice of pharmacological treatments [32]. Bladder management by patients themselves relies on good hand dexterity, powerful abdominal muscles, intact bladder sensation, and coordination of the urethral sphincter during stimulation of reflex voiding [6]. Nevertheless, the first bladder management priority in patients with chronic SCI and NLUTD should always be renal function preservation, followed by the normalization of lower urinary tract function [7].

SCI patients might have unrealistic expectations of surgical treatments for urinary incontinence, and hope to achieve both continence and self-voiding. If the treatment outcomes fall short of this, they may be disappointed, and even more so if there are undesirable side effects or complications. Thus, it is vital to ensure the patient is well informed before definitive surgical treatment. If patients cannot decide which treatment is preferable, it is best to begin with the least-invasive procedure. The patient’s agreement with the treatment modality, their motor skills, self-care capability, and levels of social, economic, and family support should also be taken into consideration.

## 5. Conclusions

Among chronic SCI patients with NLUTD and urological complications, there were high satisfaction rates with appropriate surgical procedures in terms of both the resulting improvements in urinary continence and overall bladder and voiding status. However, SCI patients frequently changed their choice of treatment, often opting for (further) surgical procedures to improve urinary continence or facilitate spontaneous voiding.

## Figures and Tables

**Table 1 jcm-11-05864-t001:** Baseline patient demographics.

	Total SCI (*n* = 700)	Cervical SCI (*n* = 238)	Thoracic SCI (*n* = 337)	Lumbar SCI (*n* = 99)	Sacral SCI Infrasacral (*n* = 26)	*p*-Value
Age of SCI (years)	35.7 ± 17.6	38.1 ± 17.4	32.2 ± 16.1	43.2 ± 18.6	28.9 ± 21.1	<0.001
Gender M/F	511/189	197/41	232/105	64/35	18/8	<0.001
Follow-up duration	13.0 ± 6.8	12.7 ± 6.6	13.2 ± 7.0	12.8 ± 6.8	13.8 ± 5.3	<0.001
Completeness of SCI	83.6%	84.0%	87.5%	72.7%	69.2	0.002
Normal hand function	73.0%	22.7%	99.1%	98.0%	100%	<0.001
VUDS findings: NDO coordinated ES NDO + DSD NDO + AD + DSD DA/DU DA/DU + ISD	51(7.3%) 375(53.6%) 85(12.1%) 162(23.1%) 26(3.7%)	10(4.2%) 156(65.5%) 46(19.3%) 25(10.5%) 1(0.4%)	26(7.7%) 203(60.2%) 36(10.7%) 57(16.9%) 14(4.2%)	10(10.1%) 15(15.2%) 1(1.0%) 69(69.7%) 4(4.0%)	5(19.2%) 1(3.8%) 2(7.7%) 11(42.3%) 7(26.9%)	0.018 <0.001 <0.001 <0.001 <0.001
Severe incontinence	296(42.3%)	86(36.1%)	174(51.6%)	24(24.2%)	12(46.2%)	<0.001
Incontinence + dysuria	262(37.4%)	92(38.7%)	116(34.4%)	46(46.5%)	8(30.8%)	0.145
Contracted bladder	223(31.9%)	80(33.6%)	115(34.1%)	23(23.2%)	5(19.2%)	0.092
Hydronephrosis	113(16.1%)	26(10.9%)	67(19.9%)	15(15.2%)	5(19.2%)	0.036
Vesicoureteral reflux	70(10.0%)	26(10.9%)	33(9.8%)	8(8.1%)	3(11.5%)	0.870
Recurrent UTI	501(71.6%)	194(81.5%)	235(69.7%)	59(59.6%)	13(50.0%)	<0.001

Abbreviations: SCI: spinal cord injury, M/F: male/female ratio, VUDS: videourodynamic study, NDO: neurogenic detrusor overactivity, ES: external sphincter, DSD: detrusor sphincter dyssynergia, AD: autonomic dysreflexia, DA: detrusor areflexia, DU: detrusor underactivity, ISD: intrinsic sphincter deficiency, UTI: urinary tract infection.

**Table 2 jcm-11-05864-t002:** Patient satisfaction with improvement of urinary continence after bladder management or urological surgeries in chronic spinal cord-injured patients with different injury levels.

	GRA	Total SCI (*n* = 700)	Cervical SCI (*n* = 238)	Thoracic SCI (*n* = 337)	Lumbar SCI (*n* = 99)	Sacral SCI Infra-Sacral (*n* = 26)	*p*-Value
Medical treatment (*n* = 145)	0, 1 2, 3	89 56(38.6%)	29 13(31.0%)	36 28(43.8%)	20 10(33.3%)	4 5(55.6%)	0.006
Detrusor Botox injection (*n* = 185)	0, 1 2, 3	35 150(81.1%)	9 62(85.9%)	21 81(78.4%)	4 5(55.6%)	1 2(66.7%)	0.000
Augmentation and CIC (*n* = 101)	0, 1 2, 3	5 96(91.4%)	0 24(100%)	3 60(95.2%)	1 10(90.9%)	1 2(66.7%)	0.001
Auto-augmentation (*n* = 5)	0, 1 2, 3	1 4(80%)	1 1(50%)	0 1(50%)	0 2(100%)	0 0	0.263
Ileal conduit diversion (*n* = 3)	0, 1 2, 3	1 2(66.7%)	0 2(100%)	1 0(0%)	0 0	0 0	0.461
Kock pouch diversion (*n* = 7)	0, 1 2, 3	0 7(100%)	0 1(100%)	0 4(100%)	0 1(100%)	0 1(100%)	0.151
Continent cystostomy (*n* = 1)	0, 1 2, 3	0 1(100%)	0 0	0 0	0 1(100%)	0 0	0.161
Suburethral sling (*n* = 14)	0, 1 2, 3	5 9(64.3%)	0 3(100%)	3 2(40%)	1 3(75%)	1 1(50%)	0.091
AUS implantation (*n* = 3)	0, 1 2, 3	1 2(66.7%)	0 0	0 1(100%)	1 1(50%)	0 0	0.297
CISC/CIC (*n* = 81)	0, 1 2, 3	56 25(30.9%)	17 6(26.1%)	23 10(30.3%)	13 6(31.6%)	3 3(50%)	0.021
Cystostomy (*n* = 100)	0, 1 2, 3	61 39(39.0%)	28 17(37.8%)	26 15(36.6%)	7 7(50%)	0 0	0.465
Indwelling urethral catheter (*n* = 55)	0,1 2, 3	40 15(27.3%)	17 8(32%)	15 7(31.8%)	6 0(0%)	2 0(0%)	0.490

GRA: global response assessment, 0: not satisfied, 1: somewhat satisfied, 2: moderate satisfaction, 3: very satisfied; SCI: spinal cord injury, CIC: clean intermittent catheterization, AUS: artificial urethral sphincter, CISC: clean intermittent self-catheterization.

**Table 3 jcm-11-05864-t003:** Patient satisfaction with improvement of urinary continence after bladder management or urological surgeries in chronic spinal cord-injured patients with different urodynamic findings.

	GRA	NDO +/− ISD (*n* = 52)	NDO + DSD (*n* = 375)	NDO + AD + DSD (*n* = 85)	DA/DU (*n* = 162)	DA/DU + ISD (*n* = 26)	*p*-Value
Medical treatment (*n* = 145)	0, 1 2, 3	7(28%) 18(72%)	49(63.6%) 28(36.4%)	4(80%) 1(20%)	20(74.1%) 7(25.9%)	9(81.8%) 2(18.2%)	0.002
Detrusor Botox injection (*n* = 185)	0, 1 2, 3	0 12(100%)	23(17.4%) 109(82.6%)	5(17.2%) 24(82.8%)	7(63.6%) 4(36.4%)	0 1(100%)	0.001
Augmentation and CIC (*n* = 101)	0, 1 2, 3	0 2(100%)	1(1.8%) 56(98.2%)	2(10.5%) 17(89.5%)	2(9.5%) 19(90.5%)	0 2(100%)	0.280
Auto-augmentation (*n* = 5)	0, 1 2, 3	0 0	1(33.3%) 2(66.7%)	0 0	0 2(100%)	0 0	1.000
Ileal conduit diversion (*n* = 3)	0, 1 2, 3	0 0	1(50%) 1(50%)	0 1(100%)	0 0	0 0	1.000
Kock pouch diversion (*n* = 7)	0, 1 2, 3	0 1(100%)	0 2(100%)	0 0	0 2(100%)	0 2(100%)	
Continent cystostomy (*n* = 1)	0, 1 2, 3	0 0	0 0	0 0	0 1(100%)	0 0	
Suburethral sling (*n* = 14)	0, 1 2, 3	0 2(100%)	1(25%) 3(75%)	0 0	1(33.3%) 2(66.7%)	3(60%) 2(40%)	0.700
AUS implantation (*n* = 3)	0, 1 2, 3	1(100%) 0	0 0	0 0	0 1(100%)	0 1(100%)	1.000
CISC/CIC (*n* = 81)	0, 1 2, 3	2(50%) 2(50%)	20(69%) 9(31%)	9(81.8%) 2(18.2%)	23(65.7%) 12(34.3%)	2(100%) 0	0.717
Cystostomy (*n* = 100)	0, 1 2, 3	0 4(100%)	31(63.3%) 18(36.7%)	7(58.3%) 5(41/7%)	22(64.7%) 12(35.3%)	1(100%) 0	0.116
Indwelling urethral catheter (*n* = 55)	0, 1 2, 3	0 1(100%)	14(70%) 6(30%)	4(50%) 4(50%)	22(88%) 3(12%)	0 1(100%)	0.018

GRA: global response assessment, 0: not satisfied, 1: somewhat satisfied, 2: moderate satisfaction, 3: very satisfied; SCI: spinal cord injury, CIC: clean intermittent catheterization, AUS: artificial urethral sphincter, CISC: clean intermittent self-catheterization, NDO: neurogenic detrusor overactivity, DSD: detrusor sphincter dyssynergia, ISD: intrinsic deficiency, DU: detrusor underactivity, DA: detrusor areflexia, AD: autonomic dysreflexia.

**Table 4 jcm-11-05864-t004:** The long-term complications of bladder management or urological surgeries for improvement of urinary continence in chronic spinal cord-injured patients.

	Total SCI (*n* = 700)	Recurrent UTI (*n* = 513)	Difficult Urination (*n* = 153)	Hydro- Nephrosis (*n* = 18)	Persistent UUI/SUI (*n* = 150)	Persistent AD (*n* = 91)
Medical treatment	145	73(50.3%)	48(33.1%)	2(1.4%)	42(29.0%)	9(6.2%)
Detrusor Botox injection	185	138(74.6%)	31(16.8%)	3(1.6%)	54(29.2%)	23(12.4%)
Augmentation and CIC	101	92(91.1%)	7(6.9%)	2(2.0%)	13(12.9%)	14(4.0%)
Auto-augmentation	5	2(40%)	1(20)	0	1(20%)	0
Ileal conduit diversion	3	2(66.7%)	0	0	0	1(33.3%)
Kock pouch diversion	7	6(85.7%)	0	1(14.3%)	1(14.3%)	0
Continent cystostomy	1	1(100%)	0	0	1(100%)	0
Suburethral sling	14	6(42.9%)	1(7.1%)	0	8(57.1%)	0
AUS implantation	3	2(66.7%)	0	0	3(100%)	0
CISC/CIC	81	57(70.4%)	29(35.8%)	4(4.9%)	17(21.0%)	8(9.9%)
Cystostomy	100	87(87%)	29(29%)	3(3%)	8(8%)	23(23%)
Indwelling urethral catheter	55	47(85.5%)	7(12.7%)	3(5.5%)	2(3.6%)	13(23.6%)

SCI: spinal cord injury, CIC: clean intermittent catheterization, AUS: artificial urethral sphincter, CISC: clean intermittent self-catheterization, UTI: urinary tract infection, UUI: urgency urinary incontinence, SUI: stress urinary incontinence, AD: autonomic dysreflexia.

**Table 5 jcm-11-05864-t005:** Overall satisfaction with final bladder and voiding condition after bladder management or urological surgeries for improvement of urinary continence in chronic spinal cord-injured patients.

	Total SCI (*n* = 700)	Very Satisfied (*n* = 338)	Satisfied but Wish to Change (*n* = 76)	Acceptable No Need to Change (*n* = 231)	Not Satisfied at All (*n* = 55)	*p*-Value
Medical treatment	145	33(49.2%)	12(8.3%)	80(55.2%)	20(13.8%)	0.000
Detrusor Botox injection	185	125(67.7%)	31(16.8%)	25(13.5%)	4(2.2%)	0.000
Augmentation and CIC	101	78(77.2%)	6(5.9%)	15(14.9%)	2(2.0%)	0.000
Auto-augmentation	5	5(100%)	0	0	0	0.191
Ileal conduit diversion	3	3(100%)	0	0	0	0.527
Kock pouch diversion	7	4(57.1%)	1(14.3%)	2(28.6%)	0	0.936
Continent cystostomy	1	1(100%)	0	0	0	0.000
Suburethral sling	14	11(78.6%)	1(7.1%)	0	2(14.2%)	0.006
AUS implantation	3	1(33.3%)	0	1(33.3%)	1(33.3%)	0.366
CISC/CIC	81	48(60.8%)	7(8.6%)	21(25.9%)	5(6.2%)	0.013
Cystostomy	100	18(18%)	12(12%)	61(61%)	9(9%)	0.000
Indwelling urethral catheter	55	11(20.0%)	6(10.9%)	26(47.3%)	12(21.8%)	0.000

SCI: spinal cord injury, CIC: clean intermittent catheterization, AUS: artificial urethral sphincter, CISC: clean intermittent self-catheterization.

## Data Availability

The data are available with the permission of Institutional Review Board after contacting with the corresponding author.

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
