# Peer review of "Satisfaction with Urinary Incontinence Treatments in Patients with Chronic Spinal Cord Injury"

_jcm, 2022, doi:10.3390/jcm11195864_

Round 1

Reviewer 1 Report

1) Electrical stimulation through neuromodulation is also known to be helpful in patients with neurogenic bladder due to spinal cord injury. Was there any case of neuromodulation in your study?

2) “After their initial treatment, patients were regularly followed up at our outpatient 93 clinic. Their renal function, bladder storage, and voiding capacity were assessed, and STI 94 tests were performed annually.”

You said “regularly followed” in the sentence above, so I hope you can describe the period a little more specifically. Also, I wonder if the renal function was confirmed only hematologically? or if other tests such as renal scans or vesicoureteral reflux and etc. was also performed, as mentioned in the introduction of materials and methods. Please tell us more about Follow-up.

3)There seems to be insufficient data on the patients demographics presented in Results. It would be better to have data on mean age or underling disease for each treatment group.

Also, adding objective data such as changes in renal functions, changes in VUR, and percentage of upper and lower recurrent UTIs.

Author Response

Dear Reviewer:

Thank you for the constructive comments. We have revised the manuscript according to your suggestions and corrections. The followings are the point-to-point replies to your individual comment.

Reviewer 1:

  • Electrical stimulation through neuromodulation is also known to be helpful in patients with neurogenic bladder due to spinal cord injury. Was there any case of neuromodulation in your study?

Reply: Thank you for the comment. Neuromodulation is surely a treatment alternative for NLUTD in patients with chronic spinal cord injury. However, this treatment has not been licensed in our country yet. We have no experience of this procedure in treating SCI patients.

2) “After their initial treatment, patients were regularly followed up at our outpatient clinic. Their renal function, bladder storage, and voiding capacity were assessed, and STI tests were performed annually.”

You said “regularly followed” in the sentence above, so I hope you can describe the period a little more specifically. Also, I wonder if the renal function was confirmed only hematologically? or if other tests such as renal scans or vesicoureteral reflux and etc. was also performed, as mentioned in the introduction of materials and methods. Please tell us more about Follow-up.

Reply: Thank you for the comment. We have added statements of screening tests and follow-up examinations. Their renal function (renal sonography, serum levels of blood urea nitrogen, creatinine, and electrolytes, estimated glomerular filtration rate, the bladder storage condition, and voiding efficiency were assessed by urodynamic study, and the tests were performed annually. (Lines 76-77, and Lines 101-103)  Patients with NDO, AD, BND, DSD, or low bladder compliance, presence of VUR received repeated VUDS every year to assess the lower and upper urinary tract conditions, any additional procedure or medications were given to protect renal function, prevent UTI, and improve urinary function.  (Lines 106-109)

  • There seems to be insufficient data on the patients demographics presented in Results. It would be better to have data on mean age or underling disease for each treatment group.

Also, adding objective data such as changes in renal functions, changes in VUR, and percentage of upper and lower recurrent UTIs.

Reply: Thank you for the comment. The baseline demographics (age, SCI completeness, VUDS characteristics, VUR, and initial bladder management) of patients with different SCI levels have been added to a new Table 1. (Line 141-142)

Reviewer 2 Report

Introduction:

-You state that NLUTD in SCI remains most difficult to manage urologic disorder.  What is your reference for this?  I would restate to say remains a challenging urologic disorder but certainly would not define as most difficult

- Lines 37-43 do not include impairment of bladder compliance which is likely the most important factor to determine upper tract deterioration and has one of highest potential to impact QoL.

- Lines 47-49 state that treatment strategies should be based on UDS data when in fact treatment should be based on combination of UDS, renal function, upper tract imaging and symptomatology. 

- Line 53 AD can be seen down to T8. 

Material & Methods

- Line 74 what is a "Renal function test" - creatinine?

- Did you use any standardized questionnaires to assess bladder symptoms and function other that the GRA

- How did you determine the order of importance of priorities for conservative vs. surgical management?  Notably there is no patient preference in any of these priorities

- How often were patients seen in clinic (more than once per year)?  How did you asses bladder storage and voiding capacity?  What is STI?  How did you determine when to repeat VUDS?

- You state patients' injuries were average at least 3 years old.  How long have these patients been followed?  

- You use the GRA scale but how did you determine baseline satisfaction? You state that 2,3 is considered satisfactory but what about patients who are happy at baseline (and thus wouldn't have improvement on the scale)? Did you ask satisfaction separately?  What did you then use to determine satisfaction?  If you use the GRA scale alone, this is a large flaw in your study data as it doesn't take into account baseline.

- How did you determine complications?  

Results:

- Your range of patient age is 1-89.  How did you include pediatric patients in your study? Arguably pediatric patients should not be included.

- How did you define AD during VUDS? You define ISD but what about patients with SUI who don't have ISD?

- You break down procedures into separate categories but many of these patient have had multimodal therapy ie history of augment currently getting intravesical botox vs. botox + sling, augment + CIC, etc.  How did you categorize these patients into 1 group?  How do you determine satisfaction of one procedure when patients have had multiple procedures and rely on multiple procedures for current continence?  

- What is your post op follow up for patients who underwent augment, diversion, etc?  Your satisfaction rates are based on GRA which is expected to be higher than prior treatments because they failed prior treatments.  The 100% satisfaction rates seem extraordinarily high for patients who have had augment, continent diversion, etc long term.

- I'm not sure that Table 2 is necessary as the VUDS is used to determine your treatment modality which would therefore affect satisfaction based on treatment response to the therapy chosen.

- How did you define recurrent UTI as a symptom?  Any evaluation for stones? Difficult urination is a broad term that could include a myriad of problems.  

Tables:

-In general the tables are very confusing to read.  You are separating patients by those who are considered to be "satisfied" vs. not satisfied.  The GRA scores on the table don't include negative scores (are you implying no patients had negative scores?).  Either separate by your 2 categories or use the entire scale.

- Table 3 should be separated into long term complications vs. reasons for dissatisfaction as complications do not necessarily lead to dissatisfaction for patients.

Author Response

Reviewer 2:

Introduction:

-You state that NLUTD in SCI remains most difficult to manage urologic disorder.  What is your reference for this?  I would restate to say remains a challenging urologic disorder but certainly would not define as most difficult

Reply: Thank you for the comment. We have revised the statement, accordingly. NLUTD in spinal cord injury (SCI) remains a challenging urological disorder. (Lines 35-36)

- Lines 37-43 do not include impairment of bladder compliance which is likely the most important factor to determine upper tract deterioration and has one of highest potential to impact QoL.

Reply: Thank you for the comment. We have added the statement of impaired bladder compliance may determine upper urinary tract deterioration. (Lines 41-42)

- Lines 47-49 state that treatment strategies should be based on UDS data when in fact treatment should be based on combination of UDS, renal function, upper tract imaging and symptomatology. 

Reply: Thank you for the comment. We have revised the statement, accordingly. “Treatment strategies should be based on the patient’s symptomatology, urodynamic data, renal function, and upper tract imaging”. (Lines 48-51)

- Line 53 AD can be seen down to T8. 

Reply: Thank you for the comment. The SCI level casing AD is usually above T6, however, as low as T8 or T10 had been reported (Spinal Cord. 43 (12): 738–40.) We have revised the SCI level causing AD to T8, accordingly. (Line 55-56)

Material & Methods

- Line 74 what is a "Renal function test" - creatinine?

Reply: Thank you for the comment. We have added the items of routine renal function test in the Method section: renal function tests (renal sonography, serum levels of blood urea nitrogen, creatinine, electrolytes, and estimated glomerular filtration rate). (Lines 76-77)

- Did you use any standardized questionnaires to assess bladder symptoms and function other that the GRA

Reply: Thank you for the comment. We did not use standardized questionnaires such as UDI-6 or IIQ-7 to assess patients’ improvement of bladder condition. Patient satisfaction with improvements in urinary incontinence as well as voiding condition was assessed using a global response assessment. (Lines 114-115)

- How did you determine the order of importance of priorities for conservative vs. surgical management?  Notably there is no patient preference in any of these priorities

Reply: Thank you for the comment. The priority of conservative versus surgical management were based on patients’ symptoms, upper tract conditions, and expectation. The procedures of bladder management were thoroughly discussed with the patients and all patients were fully acknowledged about the advantages and potential adverse events after the procedure. (Lines 85-89)

- How often were patients seen in clinic (more than once per year)?  How did you asses bladder storage and voiding capacity?  What is STI?  How did you determine when to repeat VUDS?

Reply: Thank you for the comment. Their renal function tests, the bladder storage condition, and voiding efficiency were assessed by urodynamic study, and the tests were performed every 6 months or annually depending on patient’s upper urinary tract condition. (Lines 101-103)

- You state patients' injuries were average at least 3 years old.  How long have these patients been followed? 

Reply: Thank you for the comment. The duration of the SCI being followed up from baseline was 13 ± 6.8 (range 3–25) years. (Lines 134)

- You use the GRA scale but how did you determine baseline satisfaction? You state that 2,3 is considered satisfactory but what about patients who are happy at baseline (and thus wouldn't have improvement on the scale)? Did you ask satisfaction separately?  What did you then use to determine satisfaction?  If you use the GRA scale alone, this is a large flaw in your study data as it doesn't take into account baseline.

Reply: Thank you for the comment. In this study we assessed the treatment satisfaction to the bladder management and surgical procedures for SCI patients who had urinary incontinence and received further management. If patients were satisfied to their initial management, they would not be enrolled in this study. The GRA used in this study is to assess the overall bladder storage and voiding condition after bladder management or surgery. (Lines 115-117) Patients might have improvement of urinary incontinence after treatment but were bothered by bladder emptying and rated a unsatisfactory score. (Lines 117-119)

- How did you determine complications?  

Reply: Thank you for the comment. The complication rates and treatment changes were recorded based on observational data. (Lines 124-125) Post-treatment complication was defined by having a symptom or urinary tract condition that was not occur or more severe than baseline, such as urinary incontinence, difficulty in urination, recurrent UTI, exacerbated hydronephrosis, or AD. (Lines 125-128)

Results:

- Your range of patient age is 1-89.  How did you include pediatric patients in your study? Arguably pediatric patients should not be included.

Reply: Thank you for the comment. The patients who had SCI at age 1 but was then treated in our department for 25 years. In fact, the patient was an adult at follow-up.

- How did you define AD during VUDS? You define ISD but what about patients with SUI who don't have ISD?

Reply: Thank you for the comment. AD during VUDS was defined when patients had headache, increase of spasticity, and increased blood pressure. In this study the urinary incontinence included UUI, SUI and overflow incontinence. The incontinence included patients with detrusor underactivity/areflexia and overflow incontinence without ISD. Therefore, we compared the satisfaction to bladder management or surgical procedures by different video urodynamic findings. (Table 3) (Lines 160-163)

- You break down procedures into separate categories but many of these patient have had multimodal therapy ie history of augment currently getting intravesical botox vs. botox + sling, augment + CIC, etc.  How did you categorize these patients into 1 group?  How do you determine satisfaction of one procedure when patients have had multiple procedures and rely on multiple procedures for current continence?  

Reply: Thank you for the comment. The satisfaction rated by GRA was based on the results of the initial treatment or bladder management. (Table 2,3) After the initial treatment, patients would choose another bladder management or surgical procedure to improve urinary incontinence or voiding condition. These procedures might have additive effect on urinary incontinence, reduction of adverse events, or eradicate complications. The satisfaction rate of bladder management was determined by their initial treatment procedure. Table 5 shows the overall satisfaction to final bladder and voiding condition after single or multiple procedures.

- What is your post op follow up for patients who underwent augment, diversion, etc?  Your satisfaction rates are based on GRA which is expected to be higher than prior treatments because they failed prior treatments.  The 100% satisfaction rates seem extraordinarily high for patients who have had augment, continent diversion, etc long term.

Reply: Thank you for the comment. GRA was reported based on patients’ subjective perception of improvement of urinary incontinence and voiding condition. Because the procedures of bladder management had been thoroughly discussed with the patients and all patients were fully acknowledged about the advantages and potential adverse events after the procedure. (Lines 85-89) We believe that is the cause that patient could totally accept the treatment outcome after the invasive procedures like autoaugmentation, ileal conduit, and continent cystostomy.

- I'm not sure that Table 2 is necessary as the VUDS is used to determine your treatment modality which would therefore affect satisfaction based on treatment response to the therapy chosen.

Reply: Thank you for the comment. Table 2 is reported to reveal the VUDS findings on treatment outcome. The SCI level might not parallel to the VUDS findings, this table can provide reference for the satisfactory rate after different bladder management and surgical procedures for urinary incontinence.

- How did you define recurrent UTI as a symptom? Any evaluation for stones? Difficult urination is a broad term that could include a myriad of problems.  

Reply: Thank you for the comment. We defined recurrent UTI as a complication of lower urinary tract condition. Post-treatment complication was defined by having a symptom or urinary tract condition that was not occur or more severe than baseline, such as urinary incontinence, difficulty in urination, recurrent UTI, , exacerbated hydronephrosis, or AD. (Lines 124-128) Evaluation of renal stone was included in renal function test by renal sonography. We also agree that difficult urination is a broad term that includes a myriad of problem. That is the reason why we analyze the different VUDS findings to assess the dissatisfaction rate to bladder management or surgical procedure.

Tables:

-In general the tables are very confusing to read.  You are separating patients by those who are considered to be "satisfied" vs. not satisfied.  The GRA scores on the table don't include negative scores (are you implying no patients had negative scores?).  Either separate by your 2 categories or use the entire scale.

Reply: Thank you for the comment. Although GRA with rating choices ranging from −3 to +3, to denote responses from “markedly worse” to “markedly improved”, we did not define negative GRA score in this study. A GRA score of +2 or +3 was considered a satisfactory outcome, whereas 0 or 1 was considered dissatisfied to treatment. (Lines 115-117) We could change the “GRA” item to “Satisfaction: yes or no” in the tables.

- Table 3 should be separated into long term complications vs. reasons for dissatisfaction as complications do not necessarily lead to dissatisfaction for patients.

Reply: Thank you for the comment. The new table 4 reports the long-term complications, not the cause of dissatisfaction. Patients with these complications might still satisfy to the bladder management or surgical procedures. We have revised the title of table 4. (Lines 181)

Round 2

Reviewer 1 Report

well-revised as pointed